# Structural Properties of Epoxy–Silica Barrier Coatings for Corrosion Protection of Reinforcing Steel

**DOI:** 10.3390/polym14173474

**Published:** 2022-08-25

**Authors:** Mayara Carla Uvida, Adriana de Araújo Almeida, Sandra Helena Pulcinelli, Celso Valentim Santilli, Peter Hammer

**Affiliations:** Institute of Chemistry, São Paulo State University (UNESP), Araraquara 14800-060, Brazil

**Keywords:** organic-inorganic coating, sol-gel process, epoxy–silica, corrosion protection, reinforcing steel

## Abstract

Reinforcement steel extensively applied in civil construction is susceptible to corrosion due to the carbonation process in reinforced concrete and chloride ions diffusion. Epoxy-silica-based coatings are a promising option to guarantee the long-term stability of reinforced concrete structures. In this study, the influence of the proportion between the poly (bisphenol-A-co-epichlorhydrin) resin (DGEBA) and the curing agent diethylenetriamine (DETA) on the structural, morphological, and barrier properties of epoxy–silica nanocomposites were evaluated. To simulate different stages of concrete aging, electrochemical impedance spectroscopy (EIS) assays were performed for coated samples in a 3.5 wt.% NaCl solution (pH 7) and in simulated concrete pore solutions (SCPS), which represent the hydration environment in fresh concrete (SCPS2, pH 14) and after carbonation (SCPS1, pH 8). The results showed that coatings with an intermediate DETA to DGEBA ratio of 0.4, presented the best long-term corrosion protection with a low-frequency impedance modulus of up to 3.8 GΩ cm^2^ in NaCl and SCPS1 solutions. Small-angle X-ray scattering and atomic force microscopy analysis revealed that the best performance observed for the intermediate DETA proportion is associated with the presence of larger silica nanodomains, which act as a filler in the cross-linked epoxy matrix, thus favoring the formation of an efficient diffusion barrier.

## 1. Introduction

Epoxy resins are prepolymers containing in their chemical structure the epoxy functional group, commonly referred to as oxirane, which confers reactivity to this family of materials belonging to the class of thermosetting polymers [1]. Epoxy rings can react with a variety of chemical compounds known as hardeners or curing agents, such as aliphatic or aromatic amines, and then open and rearrange to form a three-dimensional network. Glycidyl resins accounted for the largest share in the global epoxy resins market in 2020, especially bisphenol A diglycidyl ether (DGEBA) produced by reacting bisphenol A (BPA) with epichlorohydrin (IUPAC: 2-(chloromethyl) oxirane)) [1,2]. The resin is marketed in liquid form due to its versatility of application, mainly in paint, coating, sealant, and adhesive segments [2].

The combination of high mechanical and thermal stability, curability at mild temperature conditions, and good chemical resistance (acids, alkalis, and solvents) are the main factors contributing to the industrial application of epoxy resins as coatings in the aerospace, marine, automotive, and electronic industries [1]. However, the nanostructural characteristic of the cross-linked network of thermosetting polymers makes epoxy resins intrinsically susceptible to microcracking and water uptake [3,4]. These disadvantages can result in failures in the physical integrity and mechanical stability of the coatings, limiting their range of applications.

In civil construction, the use of epoxy coatings is among the methods of protecting reinforcing steel against corrosive species capable of causing the deterioration of the passive layer of steel, impairing the integrity of reinforced concrete structures (RCS) [5]. Considering the large-scale application of RCS, the importance of corrosion prevention is essential, ensuring durability, reducing maintenance and repair costs, as well reducing environmental impacts and public safety problems. However, studies have shown that pure epoxy coatings do not offer long-term protection for reinforcing steel due to water uptake and poor adhesion to the steel surface. Hence, the steel becomes susceptible to corrosion after failure in the concrete barrier, the carbonation process by CO_2_ diffusion (aging), and the diffusion of chloride ions [6]. Coatings with a thickness between 100–1000 μm typically show an impedance modulus at low frequency (|Z|_lf_) of about 10 MΩ cm^2^ in the first hours of exposure to 3.5 wt.% NaCl solution and a durability of about 40 days [7,8]. To overcome these problems, several strategies have been suggested to improve the anti-corrosion efficiency of epoxy coatings, such as the incorporation of nanoparticles, self-healing agents, and the conjugation of the polymer with an inorganic phase [9].

The incorporation of inorganic phases at the nanoscale has been used as a strategy in the development of novel coatings with elevated barrier properties [10,11]. Using this approach, epoxy–silica nanocomposites have emerged as protective coatings on a variety of metal alloys applied in different industrial sectors [12,13,14]. A key demand for the synthesis of epoxy–silica hybrids is the covalent conjugation between both phases on the nanoscale by bifunctional coupling agents such as 3-(glycidyloxypropyl)trimethoxysilane (GPTMS) or (3-aminopropyl)triethoxysilane (APTES). In these molecules, the siloxane functional group interacts with silica, and the epoxy or amino groups interact in the presence of a curing agent with the epoxy resin [12,13,14,15,16]. Results of recent studies confirmed that composites that combine the epoxy phase with nanometric silica nodes, derived from sol-gel polycondensation products of tetraethylorthosilicate (TEOS), contributed to improved mechanical, adhesive, and anti-corrosion properties of the coatings [12,13,14,15,16].

Chen et al. prepared hybrid nanocomposites based on epoxy resin cured with APTES using different proportions of TEOS. The results showed that adding 4 wt.% TEOS led to the formation of a homogeneous hybrid structure with improved compatibility between the epoxy and the silica phase, a higher crosslink density of the hybrid network, and an improved adhesion to AA2024 aluminum alloy substrate by the formation of the Si−O−Al bonds [14]. In another study, Bakhshandeh et al. correlated the anticorrosive efficiency with the incorporation of silica (TEOS) in hybrid nanocomposites based on DGEBA resin cured with APTES. With a coating thickness of 140 μm on steel (ST 37), the authors obtained |Z|_lf_ values of approximately 1 GΩ cm^2^ after 45 days of immersion in a 3.5 wt.% NaCl solution [12]. Torrico et al. studied the influence of the proportion between TEOS and GPTMS on the structural and anticorrosive properties of epoxy–silica nanocomposites obtained from the curing of DGEBA resin with diethylenetriamine (DETA). The results showed that an intermediate TEOS to GPTMS ratio of 1.5 favored the formation of a highly condensed silica phase covalently linked to the epoxy matrix, playing a key role in the high corrosion resistance (0.3 GΩ cm^2^) of coatings on A1020 carbon steel immersed for 42 days in a 3.5 wt.% NaCl solution [13].

Most studies focused on the optimization of the epoxy–silica properties by evaluating the influence of varying silica content using different coupling agents [12,13,14,15,16]. However, the high connectivity of the hybrid structure depends on the sensitive balance of precursor proportions. Factors such as the chemical structures of the resin and hardener, resin/hardener ratio, and type of the coupling agent determine the final structure. In addition, the application of epoxy–silica hybrids as coatings for the corrosion protection of reinforcing steel is poorly explored, especially in solutions that simulate the environment of concrete before and after the carbonation process.

Given the gap, this study sought to evaluate the influence of the epoxy resin/hardener ratio on the properties of DGEBA-GPTMS-TEOS-based epoxy–silica hybrids as potential coatings for the corrosion protection of reinforcing steel. Epoxy–silica hybrids prepared at different DETA/DGEBA ratios had their chemical and morphological structure characterized by a wide range of techniques. The anti-corrosion efficiency of the coatings on reinforcing steel was evaluated by electrochemical impedance spectroscopy (EIS) in a neutral 3.5 wt.% NaCl, carbonated, and alkaline-simulated concrete pore solutions (SCPS).

## 2. Materials and Methods

### 2.1. Hybrid Synthesis and Coating Deposition

Epoxy–silica hybrids were synthesized using poly (bisphenol-A-co-epichlorhydrin) (DGEBA, Mn = 377 g/mol, Sigma-Aldrich, St. Louis, MO, USA) as epoxy resin, diethylenetriamine (DETA, Sigma-Aldrich) as the curing agent, and the silanes (3-glycidoxypropyl) trimethoxysilane (GPTMS, Sigma-Aldrich) and tetraethylorthosilicate (TEOS, Sigma-Aldrich) as the coupling agent and silica source, respectively.

The synthesis of the epoxy–silica hybrids was carried out in two steps. The first step involved the curing reaction of the epoxy rings in the DGEBA and GPTMS structures by DETA using tetrahydrofuran (THF, Sigma Aldrich) as a solvent in a reflux system under constant stirring for 4 h at 70 °C. In the second step, the silica nanophase was synthesized in a closed flask from the sol-gel reactions of hydrolysis and condensation of GPTMS and TEOS using an aqueous HNO_3_ solution (Sigma Aldrich, pH 1) and ethanol (Sigma Aldrich) as solvent, during 1 h at 25 °C at constant stirring. Finally, the solutions obtained individually were mixed for 5 min at 70 °C to obtain a homogeneous hybrid solution. The molar ratios between the reagents and the volumes used were: DGEBA/GPTMS/TEOS = 1/1/1 (2.58 mL/1.76 mL/1.77 mL), Ethanol/H_2_O = 0.5 (1.23 mL/0.75 mL) and H_2_O/Si = 3.5. Three DETA/DGEBA molar ratios were studied: 0.3 (0.26 mL), 0.4 (0.34 mL), and 0.6 (0.52 mL), referring to the samples as D0.3, D0.4, and D0.6.

To obtain epoxy–silica-coated samples, the steel substrates (Ø 25 mm × 2 mm) were immersed three times into the hybrid solution using a dip-coater (Microtube, Araraquara, Brazil) at a controlled immersion/emersion speed of 140 mm min^−1^, allowing a drying time of 10 min between each immersion. The remaining hybrid solution was poured into Teflon cups and heat-treated together with the coated substrates at 60 °C for 48 h and 100 °C for 3 h, to ensure complete curing and polycondensation of the organic and inorganic phases and elimination of solvents. The representation of the experimental procedure for the preparation of epoxy–silica hybrids is shown in Appendix A.

Reinforcing steel substrates with nominal chemical composition of C = 0.35%; Si = 0.50%; Mn = 1.50%; P = 0.05%; S = 0.05% and CE (Carbon equivalent) = 0.55% were obtained by cutting commercially purchased bars (ArcelorMittal, Piracicaba, Brazil). Before deposition, the substrate surfaces were prepared by polishing in a polisher (Arotec, Cotia, Brazil) using water sandpapers in 320, 600, and 1500 grain sizes. Subsequently, the substrates were placed in a cup filled with isopropanol for cleaning in an ultrasonic bath for 10 min and then dried under nitrogen flow.

### 2.2. Characterization Techniques

#### 2.2.1. Structural and Thermal Properties

The chemical structure of the hybrid coatings was evaluated using a Hyperion 2000 Fourier transform infrared microscope (Bruker, Billerica, MA, USA) in the attenuated total reflection mode (ATR-FTIR). Spectra acquisition was performed in the range of 4000 to 400 cm^−1^ with a resolution of 4 cm^−1^ and 32 scans. The structural influence after exposure to 3.5 wt.% NaCl and simulated concrete pore solutions was evaluated for freestanding hybrid films after 30 days of immersion.

The thermal stability and degradation events of the freestanding hybrids were evaluated by thermogravimetry (TG) using SDT Q600 equipment (TA Instruments, New Castle, DE, USA) for the freestanding films. The analyses were performed using 7 mg of the sample in nitrogen and synthetic air atmosphere, both with a continuous flow rate of 100 mL min^−1^, a heating rate of 5 °C min^−1^, and the temperature range from 30 to 700 °C.

The residual porosity of the freestanding hybrid films was analyzed by helium (AccuPycc 1330, Micromeritics, Norcross, GA, USA) and Dry Fluid (GeoPyc 1360, Micromeritics) pycnometry measurements. The porosity percentage (*P*) was calculated considering the parameters real skeletal volume (*V_Sk_*) and bulk apparent volume (*V_B_*) using the relation $P=\left(V_{B}-V_{Sk}/V_{B}\right)\times 100$.

The size and form of silica nanodomains in the epoxy matrix were studied using small-angle X-ray scattering (SAXS). The scattering profiles were obtained for the freestanding hybrid films using the Nano-In Xider equipment (Xenocs, Grenoble, France) with a perpendicularly focused Cu Kα (*λ* = 1.54 Å) radiation source and a 2D Dectris Pilatus 3 detector at 938 mm distance from the sample. The scattering intensity *I(q)* is a function of the scattering angle (*2θ*), and scattering vector (*q*) is given by the relation *q = (4 π/λ) sen (2θ)*, where *λ* is X-ray wavelength. In hybrid nanocomposites, the scattering is a result of a contrast between the electronic density of silica nanodomains and the polymer matrix. The SAXS profiles were fitted using the SASView software [17] with the Guinier–Porod function proposed by Hammouda [18] for determining the size and form of the scattering objects:(1)$$I\left(q\right)=\frac{G}{q^{s}}e^{\left(\frac{-q^{2}R_{g}^{2}}{3-s}\right)}\;q\leq q_{1}$$
(2)$$I\left(q\right)=\frac{D}{q^{\alpha }}\;q\geq q_{1}$$

*I(q)* is the scattering intensity, *q* is the scattering vector modulus, *R_g_* is the gyrating radius, α is the Porod exponent, and *G* and *D* are the Guinier and Porod scale factors, respectively. Guinier’s expression is used for *q ≤ q_1_* and Porod’s expression is used for *q ≥ q*_1_, with *q*_1_ = 0.53 nm^−1^, as shown in Equations (2) and (3). The parameter *s* is used to model non-spherical objects, where *s* = 0 for 3D globular objects (spheres); *s* = 1 for 2D objects (such as rods); and *s* = 2 for 1D objects (such as lamellae or platelets), considering a dimensionality parameter (3 − *s*) is set 3 for spherical objects, 2 for rods, and 1 for lamellae or platelets. The gyration radius for a sphere of radius *R* is given by $R_{g}=R\sqrt{\left(3/5\right)}$, for a randomly oriented cylinder of radius R is $R_{g}=R/\sqrt{2}$, and for a randomly oriented lamella of thickness *T*, $R_{g}=T/\sqrt{12}$.

X-ray photoelectron spectroscopy (XPS) was applied to determine the elemental composition of the hybrids and to investigate the chemical bonding structure by the deconvolution of the carbon (C 1s), oxygen (O 1s), nitrogen (N 1s), and silicon (Si 2p) spectra. The experiments were carried out at a pressure of less than 10^−7^ Pa using a commercial UNISPECS UHV Surface Analysis spectrometer (SPECS, Berlin, Germany). The Mg Kα line was used (hν = 1256.6 eV) and the pass energy for the high-resolution spectra was set to 15 eV. The inelastic background of the spectra was subtracted using Shirley’s method. The composition of the near surface region was determined with an accuracy of ±5% from the ratio of the relative peak areas, corrected by Scofield’s sensitivity factors of the corresponding elements. The spectra were fitted by multiple Voigt profiles without placing constraints using the CasaXPS software (Casa Software Ltd., Teigmouth, UK).

#### 2.2.2. Thickness and Surface Properties

The average thickness of the hybrid coatings was obtained from 10 measurements performed on a F3-CS spectrometer (Filmetrics, San Diego, CA, USA). The uniformity of the hybrid coatings was evaluated using an XJM900 optical microscope (Kozo).

Surface topography maps were obtained by atomic force microscopy (AFM) in tapping mode using a Technologies 5500 microscope (Agilent, Santa Clara, CA, USA), in three different regions with an area of 2 μm^2^. The root mean square (RMS) roughness (*R_RMS_*) of the coatings was determined by analyzing the images using Gwyddion software.

The surface wettability for the hybrid coatings was determined by contact angle measurements. The analyses were performed using contact angle optical equipment (Dataphysics, Filderstadt, Germany) with an attached camera and SCA20.2.0 software. The average contact angle value between the coating and the 7 μL distilled water drop was obtained from five measurements on each sample.

The adhesion of the coatings on the steel substrate was evaluated by pull-off adhesion tests using an Automatic F510-20T gauge (Elcometer, Manchester, UK). To perform the test, the samples were prepared by gently scratching the surface of the coatings with SiC sandpaper grit 600, cleaned with isopropanol, the aluminum dolly (Ø 10 mm) was fixed perpendicularly with Araldite^®^ 2000+ glue, and then heat treated in an oven at 100 °C for 3 h to cure the glue. After 24 h, the samples were tested by applying tensile force at a constant rate of 0.8 MPa s^−1^ until the coatings were detached. The critical detachment force (MPa) was measured in duplicate for each sample.

#### 2.2.3. Corrosion Protection

The anticorrosive efficiency of the hybrid coatings in comparison to the uncoated steel substrate was evaluated by electrochemical impedance spectroscopy (EIS) using the Reference 600 potentiostat (Gamry, Warminster, PA, USA). The measurements were performed at room temperature (25 °C) using an electrochemical cell with: (i) uncoated and coated reinforcing steel substrate as the working electrodes; (ii) Ag|AgCl|KClsat or Hg/HgO reference electrodes; and (iii) a platinum counter electrode connected by 0.1 μF capacitor to a platinum wire. The following solutions were used to evaluate different corrosive environments: 3.5 wt.% NaCl, carbonated (SCPS1: pH 8 containing Na_2_SO_4_, CaCO_3_, and NaCl), and alkaline (SCPS2: pH 14 containing Ca(OH)_2_, NaOH, KOH, and Na_2_SO_4_) simulated concrete pore solutions (SCPS). The composition of the SCPS in detail is shown in Table 1. The Hg/HgO electrode was used in the SCPS2 solution due to the occurrence of potential fluctuations of the Ag/AgCl electrode in alkaline environments [19]. After reading and stabilizing the open circuit potential (OCP), a sinusoidal potential disturbance of 10 mV_rms_ was applied for frequencies in the range of 1 MHz and 4 mHz. All samples were measured in duplicate.

To obtain electrochemical parameters, the impedance curves were fitted with equivalent electrical circuits (EEC) using Zview^®^ software (Scribner Associates, Southern Pines, NC, USA). The deviation from the capacitive response of an ideal capacitor was taken into account by incorporation of the constant phase element (CPE), and the effective capacitance was calculated using Equation (3) [20], which contains parameters *Q* and *n* of the CPE and the resistance *R* of the coating.
(3)$$C=Q^{1/n\;\;}R^{(1-n)/n\;\;}$$

Water uptake ϕ(%) was calculated using the Brasher–Kingsbury relation [21] given by Equation (4), which considers the high-frequency coating capacitances *C_t_* and *C_0_* obtained by EEC fitting for time t and initial of exposure (*t* = 2 h), and the dielectric constant of water *ε_w_* equal to 78.3 at 25 °C [22].
(4)$$\phi \left(\%\right)=\frac{\log\left(C_{t}/C_{0}\right)}{\log\varepsilon_{w}}\times 100$$

## 3. Results and Discussion

The chemical structure of the epoxy–silica hybrids was evaluated by the identification of vibrational bands of the FTIR spectra and comparison with spectra of precursor reagents (Figure 1, Table 2). The contribution of the organic phase was confirmed by the presence of vibrational bands that are characteristic of functional groups of the DGEBA, DETA, and GPTMS precursors, such as vibrations of C−H stretching of CH_2_, CH_3,_ and CH aromatic and aliphatic groups in the range of 2995–2850 cm^−1^ [23,24]; stretching vibrations of the C=C bond of the aromatic ring at 1610 cm^−1^ and the C−C bond at 1508 cm^−1^ for DGEBA resin [24]; overlapping bands that refer to C=C vibrations of aromatic rings and N−H bond of primary amine at 1578 cm^−1^ [24,25]; bending vibrations of CH_2_ groups at 1458 cm^−1^ [24]; C−C−O−C group vibrations at 1235 cm^−1^ and 1180 cm^−1^ [26,27]; C−O−C stretching vibrations of the ether group at 1035 cm^−1^, indicating the opening of the epoxy rings in the presence of amine groups [25]; and the 1,4-aromatic ring substitution (−C=C<) for DGEBA epoxy resin is evidenced by the presence of a band at 825 cm^−1^ [28]. The appearance of the band at 1722 cm^−1^ can be attributed to the formation of carboxyl groups (C=O) from the thermal oxidation of CH_2_ groups in the α position to oxygenated functional groups during the heat treatment step (100 °C) [29].

The curing reaction of the epoxy resin in the presence of the DETA was evidenced by the disappearance of the band at 915 cm^−1^ associated with the C−O−C bond of the oxirane ring of the DGEBA resin and the GPTMS coupling agent [24]. An additional feature of the curing reaction is the appearance of a peak at 1661 cm^−1^ of stretching vibrations of the N−H groups of primary amines, which exhibits a slight increase in intensity for higher DETA/DGEBA ratios (inset Figure 1) [24]. Please note that at a DETA fraction of 0.4, there is one epoxide ring for each hydrogen atom of the amine group in a stoichiometric reaction.

The formation of the silica phase by the sol-gel process from the precursors GPTMS and TEOS can be identified by the presence of characteristic vibrational bands in the region 1100−400 cm^−1^ (Figure 1, Table 2); however, the overlapping vibrations of functional groups of the precursor reagents make it difficult to assign the stretching bands of the Si−O−Si bonds at 1180 cm^−1^ and 1083 cm^−1^ [23]. In addition to these bands, other vibrational bands characteristic of Si−O−Si bond formation were evidenced in the region of 800–400 cm^−1^, such as symmetric stretching of the Si−O bond at 806 cm^−1^ and 430 cm^−1^, and stretching of the Si-O bond at 557 cm^−1^ indicating the formation of 4-membered siloxane structures ((SiO)_4_) [23]. In the region of 3600–3080 cm^−1^, an overlap of vibrational stretching bands of the N−H bond of primary and secondary amines [27] and the stretching band of hydroxyl −OH groups can be observed, the latter coming from the curing reaction of epoxy ring opening, residual solvents from the synthesis (water and ethanol), and/or non-polycondensed silanols (Si−OH) groups [24,28].

Information on the nanostructure of the epoxy–silica hybrids prepared at different DETA/DGEBA ratios was obtained from analyses of small angle X-ray scattering (SAXS) profiles (Figure 2). The scattering profiles obtained for the samples were fitted using the Guinier–Porod model proposed by Hammouda [18]. The model enables the extraction of the gyration radius (*R_g_*) related to the root-mean-square distance of a silica particle from the center of its electronic density [30], the dimension variable (*s*) referring to the shape, and the Porod coefficient (α) associated with the geometry of the silica scattering nanodomains (Table 3). The presence of a broad knee in a *q* range between 0.4 and 3 nm^−1^ is similar to that already observed for epoxy–silica hybrids and can be attributed to the scattering of silica nanodomains dispersed uniformly in the epoxy matrix [31,32]. Matějka et al. obtained epoxy–silica hybrids by a two-step synthesis, combining hydrolysis of TEOS with the formation of the epoxy network in the presence of the curing agent [33]. According to the authors, the growth of silica nanodomains with a size between 10 and 20 nm is sterically restricted by the rigid epoxy matrix, which hinders the densification of the formed structures during the 100 °C drying step. 

From the data analysis, *R_g_* values close to 10 nm were obtained (Table 3). The Porod coefficients α between 2.7 and 3.0 suggest the presence of silica nanodomains with a rough interfacial structure, and the dimension variable with values between 0.28 < *s* < 0.48 indicates the presence of 3D scattering objects close to spheres (*s* = 0). A deviation from Porod’s law associated with electron density fluctuations is observed for all samples at higher *q*–values (*q* > 3 nm^−1^), which may be associated with the presence of smaller objects, interfacial roughness, defects, or microporosity of the epoxy matrix at the molecular scale [34,35].

A slightly higher *R_g_* value was found for the D0.4 sample, indicating the formation of larger silica nanodomains. Combining the results obtained by FTIR, in particular the appearance of the band at 557 cm^−1^, it is possible to suggest that the silica nanodomains are formed of 4-membered siloxane structures ((SiO)_4_) [23,32]. Furthermore, as the largest silica domains were formed for the intermediate DETA/DGEBA ratio, there is no clear relationship between the size of silica nanodomains and the crosslinking density of the epoxy resin by adding larger amounts of DETA. 

X-ray photoelectron spectroscopy (XPS) was applied to study the influence of increasing DETA addition on the local bonding environment and composition of the epoxy–silica coatings. The quantitative analysis confirmed the formation of the hybrid phase, as well as the increasing quantity of nitrogen in the samples (Figure 3). Within the experimental error of the technique (±5%), the elemental composition of the coatings is in agreement with the nominal molar proportions of polycondensed and polymerized precursors with DETA/DGEBA/GPTMS/TEOS proportions of (0.3, 0.4, 0.6)/1/1/1, corresponding to about 74 at.% carbon, 18 at.% oxygen, 5 at.% silicon, and nitrogen content of 1.8 at.% (D0.3), 3.4 at.% (D0.4), and 4.9 at.% (D0.6).

The deconvolution of the C 1s, Si 2p, and N 1s spectra confirmed the presence of the main functional groups of the hybrid (Figure 2b). The fitted C 1s components correspond to the three bonding environments of C−C bonds in DGEBA at 284.6 eV, C−N bonds of DETA at 285.7 eV, and C−O/C−OH bonds of DGEBA and GPTMS at 286.7 eV. The increment of the C−N signal with increasing DETA fraction is evident. The weak signal of the ester group (O−C=O) at 288.5 eV may be due to contamination of the samples by adventitious carbon, which contributes also to a low extent of C−O and C−C components. As expected for the proportion TEOS/GPTMS = 1, the Si 2p peak shows for all samples the presence of two components of equal intensity referring to polycondensation products of TEOS (SiO_2_ at 103.5 eV) and GPTMS (C−SiO_x_ at 102.6 eV). The N 1s spectra were fitted with three components assigned to C−N−C bonds at 399.0 eV of cured epoxy, residual primary and secondary amines at 399.9 eV, and a small contribution of protonated nitrogen (−NH_3_^+^) at 402.0 eV, most probably due to the interaction of secondary amines with the silanol group of hydrolyzed GPTMS [36]. With increasing DETA fraction, the sub-peak intensity of residual amines increases, indicating that the primary amine of DETA and excess of secondary amines (D0.6) do not contribute to the curing reaction of the reticulated network. 

The thermal degradation of the epoxy–silica hybrids was studied by thermogravimetric (TG) analysis in nitrogen and air atmospheres (Figure 4, Table 4). The value for the onset temperature of thermal stability (T_0_) was obtained for a 10% mass loss. This percentage was chosen due to an initial dehydration process at approximately 100 °C. The hybrids exhibited thermal stability in a nitrogen atmosphere between 254–328 °C, with the highest value obtained for the D0.4 sample. The TG and DTG (first derivative of TG) curves in the N_2_ atmosphere (Figure 4a) show four degradation events, which are associated with the thermal decomposition of the organic and inorganic phases [13,37]. The first event (T_1_) around 150 °C is related to dehydration by the elimination of hydroxyl groups and solvent molecules. The second event (T_2_) at about 370 °C refers to the breaking of C−C and C−O bonds of the polymeric structure, and the third event (T_3_) at approximately 420 °C is related to the breaking of the stronger N−C bonds formed during the curing reaction of DGEBA epoxy resin with DETA. A strongly suppressed fourth degradation event (T_4_) at approximately 520 °C is associated with SiO_2_ formation by the loss of the silanol groups in the form of SiO_x_(OH)_y_ species. The proposed thermal decomposition mechanism for the epoxy–silica hybrid in a nitrogen atmosphere is shown in Figure 4c.

A complex thermal degradation process is observed in an oxidizing atmosphere (Figure 4b), resulting in a complete thermo-oxidative decomposition of the epoxy resin forming a variety of volatile products (acetone, alcohols, water, phenolic compounds, etc.), and the release of gases such as propylene, carbon monoxide, and carbon dioxide [38,39]. The residue formed after complete degradation in an oxidative atmosphere is composed purely of the inorganic silica phase. The percentages of residues in the N_2_ atmosphere are approximately twice as high as in air (Table 4), an indication that in an inert atmosphere a large amount of aromatic carbon residue (coke and carbonaceous char) is formed, as a result of the partial decomposition of the phenolic epoxy structure [40]. The thermal decomposition events in N_2_ and the residue percentages are shown in Table 4.

The epoxy–silica hybrid coatings, deposited on reinforcing steel, are transparent, colorless, and homogeneous, as can be seen in Figure 5a for the coated sample and monolith, as well as the optical micrographs (Figure 5b). The helium and solid-fluid pycnometry results showed for the epoxy–silica hybrids a residual porosity of less than 4% (Table 3) after thermal treatment, which may influence the anti-corrosion efficiency of the coatings in the presence of electrolytes. These porosity values agree with the deviation of Porod’s law observed by SAXS.

AFM topographical maps of the epoxy–silica hybrids, shown in Figure 5c, were used to extract the R_RMS_ roughness of the coatings, showing values between 0.6 and 1.9 nm (Table 3). The presence of larger silica domains obtained by SAXS for the D0.4 coating and its higher surface roughness (1.9 nm) can be correlated to the water contact angle results, which show for this sample the highest value of 88.7°, on the threshold to hydrophobicity. On the other hand, the D0.6 coating shows the smoothest surface, a result in agreement with the smaller silica domains suggested by SAXS results. This coating and D0.3 sample showed contact angles close to 80° (Table 3), indicating a slightly less hydrophilic surface than the pure DGEBA epoxy coatings of about 70° [41].

The evaluation of the anticorrosive efficiency of epoxy–silica with different DETA/DGEBA ratios was performed by electrochemical impedance spectroscopy (EIS) assays in neutral 3.5% NaCl solution and simulated concrete pore solutions, SCPS1 (pH 8) and SCPS2 (pH 14), corresponding to different concrete environments. The impedance modulus values at 4 mHz (|Z|_lf_) and the phase angle profiles were used to evaluate the corrosion resistance of the coatings (Table 5). To exclude a possible influence of the coating thickness on the anticorrosive performance, the recorded |Z|_lf_ values after 1 day of exposure were normalized by the average coating thickness of 10 µm (Table 5). Figure 6 and Appendix A (duplicate) display the Bode plots of the samples after 1 day of exposure to neutral saline and alkaline solutions. The comparison with uncoated reinforcing steel shows that the D0.4 coating provides excellent protection in saline and SCPS1 solutions with a |Z|_lf_ value of up to 4 GΩ cm^2^ and phase angle values close to −90° over a wide frequency range. At harsh conditions of pH 14, the profiles show that the coatings are affected by the electrolyte, presenting in the mid-frequency a phase angle depression, known as an anomaly, indicating the beginning formation of percolation paths between the electrolyte and the coating/metal interface [10,11]. The use of a lower proportion of curing agents as in sample D0.3 resulted in poorly performing coatings in alkaline solutions. Although coatings prepared at a higher DETA/DGEBA ratio (D0.6) had a higher thickness (15.2 μm) and slightly lower residual porosity, a significant performance improvement is not evident from the data. Additionally, Figure 6c shows that after exposure to an alkaline medium, improved passivation of the bare steel took place. At a pH of 14, the impedance modulus increased about 2 orders of magnitude, as predicted by the Pourbaix diagram for the Fe/H_2_O system [6].

Table 5 shows that epoxy–silica coating with the intermediate DETA/DGEBA ratio (D0.4) presents the best long-term stability in all electrolyte solutions, emphasizing the importance of optimizing the coating formulation, in this case, the proportion of the curing agent to achieve increased network connectivity. Therefore, the D0.4 coating was chosen to study in more detail the long-term performance. Although the Bode plots of Figure 7 and Appendix A show a gradual decrease in the impedance modulus and an upshift of the breaking point frequency (phase angle of −45°) during long-term immersion in 3.5 NaCl and SCPS1 solution, excellent coating durability of up to 330 days was achieved. The drop in performance may be related to the residual porosity, detected by pycnometry, which is characteristic of the cross-linked structure of epoxy resins resulting in the formation of water uptake paths toward the coating/steel interface.

The EIS curves obtained for sample D0.4 (Figure 7a–c) were fitted using the equivalent electrical circuits (EEC) shown in Figure 7d–f for the first and last day of immersion in the 3.5 wt.% NaCl and simulated concrete pore solutions (pH8 and pH 14). The chi-square (*χ^2^*) values in the range 10^−3^–10^−4^ indicate good fit quality for the extraction of electrochemical parameters presented in Table 6. The three EECs used for fitting the EIS curves were composed of the solution resistance (*R_s_*) connected in series with RC time constants, using the constant phase elements as pseudo-capacitances. The EIS curves obtained for the D0.4 samples exposed to 3.5 wt.% NaCl during 2 h and 240 days and SCPS1 (2 h), were fitted using an EEC composed of two-time constants, *R_1_*/*CPE_1_* and *R_2_*/*CPE_2_* (Figure 7d), associated with the resistance and capacitance of the near-surface layer and at the coating/steel interface, respectively. For the SCPS1 solution after 330 days and SCPS2 after 2 h, the degradation of the barrier property of the coating was evidenced by the presence of a third-time constant related to the charge transfer resistance (*R_ct_*) and the double layer capacitance (*CPE_dl_*) associated with the corrosive activity of the metal surface (Figure 7e). After 7 days of exposure to SCPS2 solution (pH 14), the curve profile is similar to the uncoated reinforcing steel (Figure 6c) fitted with one time constant (*R_ct_/CPE_dl_*), indicating coating degradation in the highly alkaline medium (Figure 7f). 

In neutral saline and SCPS1 solution, the D0.4 coatings showed an initial *R_2_* resistance of up to 1.29 GΩ cm^2^ and capacitance value < 0.75 nF cm^−2^ (Table 2). After prolonged exposure (NaCl for 240 days and SCPS1 for 330 days), the initial coating resistances, R_1_ and R_2_, undergo a drop due to the electrolyte permeation. The water uptake, φ(%), calculated for D0.4 coating using the Brasher–Kingsbury relation (Equation (4)) [21], was 3.63% in NaCl and 9.28 % in SCPS1. In the cases of SCPS2 solution, no water uptake values were obtained after 7 days due to the degradation of the coating by the hydrolysis reactions of silica and epoxy matrix in alkaline pH.

Structural changes of D0.4 freestanding films were evaluated using FTIR after exposure to the electrolytes (3.5 wt.% NaCl, SCPS1, and SCPS2) after 30 days. Figure 7 shows expressive changes in the absorption bands after exposure to the SCPS2 solution (pH 14) compared to the intact sample and those in contact with 3.5 wt.% NaCl and SCPS1 solution. The absorption bands in different spectral regions, shown in the inset of Figure 8, evidence the occurrence of water uptake, and changes in the chemical structure of DGEBA and the silica nanophase (Figure 8). After exposure to SCPS2, a strong intensity increase of the band in the range of 3600–3000 cm^−1^ is observed, associated with a stretching of OH groups due to hydrolysis reaction and water uptake [42,43]. Water uptake is also evidenced by the increment of the band at 1645 cm^−1^, overlapping with the stretching vibration of the N−H bond of amine groups [43]. The ether bonds of the DGEBA structure are susceptible to hydrolysis reactions under the formation of phenol groups [42]. The disappearance of the bands at 1722 cm^−1^ and 925 cm^−1^ indicates the hydrolysis reactions of the C=O and C−O groups, respectively. Additionally, a decrease in the band at 1080 cm^−1^ indicates hydrolysis-induced breaking of the C−N bond formed during the curing reaction of DETA with the DGEBA resin and GTPMS [44]. The oxidation of the epoxy matrix in the presence of water and oxygen is evidenced by the increase in the intensity of the band at 1180 cm^−1^ associated with the C−C−O−C bond vibrations. The disappearance of vibrational bands at 557 cm^−1^ and 430 cm^−1^ characteristic of the Si−O bonds in the silica nanophase [23] and the appearance of a band at 960 cm^−1^ associated with the Si−OH bond stretching vibrations [10,23] indicate hydrolysis of the Si−O−Si bonds. Thus, the FTIR results obtained for the sample in highly alkaline media show that hydrolysis of functional groups of the epoxy matrix and Si−O−Si bonds contribute to the water uptake (Table 6) and finally lead to the failure of the coating, as observed in Figure 7c.

## 4. Conclusions

Epoxy–silica hybrids were obtained by combining the curing reactions and sol-gel process using different ratios of curing agent and epoxy resin (DETA/DGEBA), in the range of 0.3–0.6, and deposited on a reinforcing steel substrate. Coatings with a thickness of a few micrometers exhibit a smooth and homogeneous surface and high adhesion to the reinforcing steel substrate of up to 16.1 MPa. 

FTIR and XPS analyses confirmed the formation of the epoxy–silica hybrid structure.

Thermal and EIS analyses showed that an intermediate DETA to DGEBA molar ratio of 0.4 resulted in hybrid coatings with modified nanostructure that provides a high thermal stability (328 °C) and excellent long-term anticorrosive performance in saline and alkaline (pH 8) medium. This was confirmed by SAXS and AFM results revealing the presence of a uniform distribution of silica domains with a size of 11.3 nm covalently linked to the epoxy matrix, resulting in a surface roughness of 1.9 nm and a contact angle of 88.7°.

Long-term EIS assays of D0.4 coatings in 3.5 wt.% NaCl and SCPS1 (pH 8) solutions showed excellent durability of 330 days maintaining a high corrosion resistance, with a low-frequency impedance modulus of up to 3.8 GΩ cm^2^. The improved corrosion performance of the D0.4 coatings can be related to the presence of lager silica nanodomains filling the cross-linked structure of the epoxy resin, thus acting as an efficient barrier against electrolyte permeation. A similar initial performance was observed in the SCPS2 solution (pH 14), however, with shorter durability due to coating degradation induced by hydrolysis reactions of DGEBA resin and the silica nanophase in a harsh environment. 

The results presented here highlight the potential of epoxy–silica coatings for the protection of reinforcing steel in aged carbonated (pH 8) concrete structures and point to their limitations and harsh (pH 14) alkaline electrolytes.

## Figures and Tables

**Figure 1 polymers-14-03474-f001:**
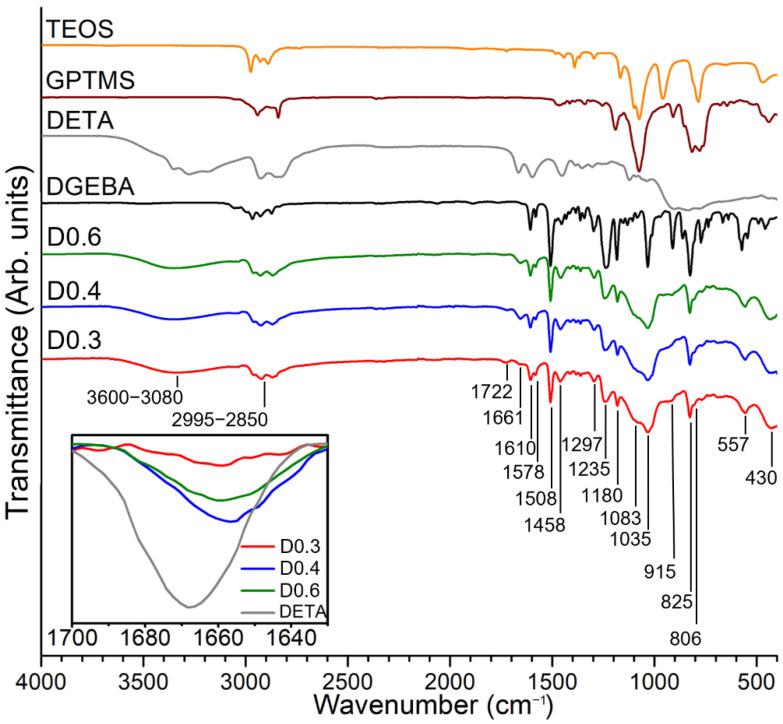
FTIR spectra of the precursor reagents (DGEBA, DETA, GPTMS, and TEOS) and epoxy–silica hybrids prepared with different curing agent/epoxy resin ratios (D0.3, D0.4, and D0.6). Insert: Band at 1661 cm^−1^ associated with the N–H bond stretching vibrations of epoxy–silica hybrids compared to the DETA precursor.

**Figure 2 polymers-14-03474-f002:**
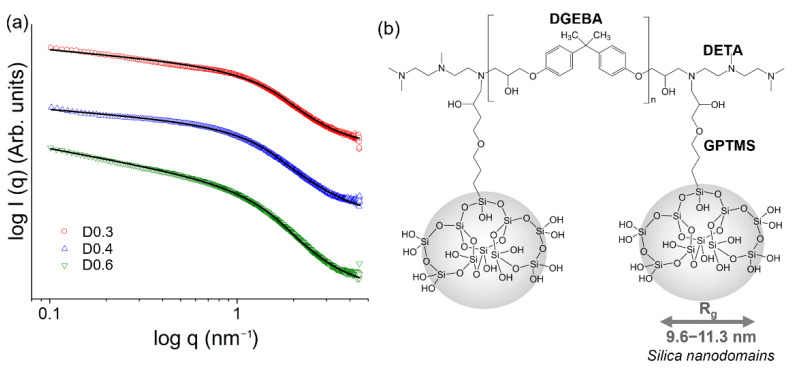
(**a**) SAXS scattering profiles of epoxy–silica hybrids prepared with different proportions of curing agent (DETA) fitted according to the Guinier–Porod model (black lines), (**b**) Structural representation of epoxy–silica hybrids.

**Figure 3 polymers-14-03474-f003:**
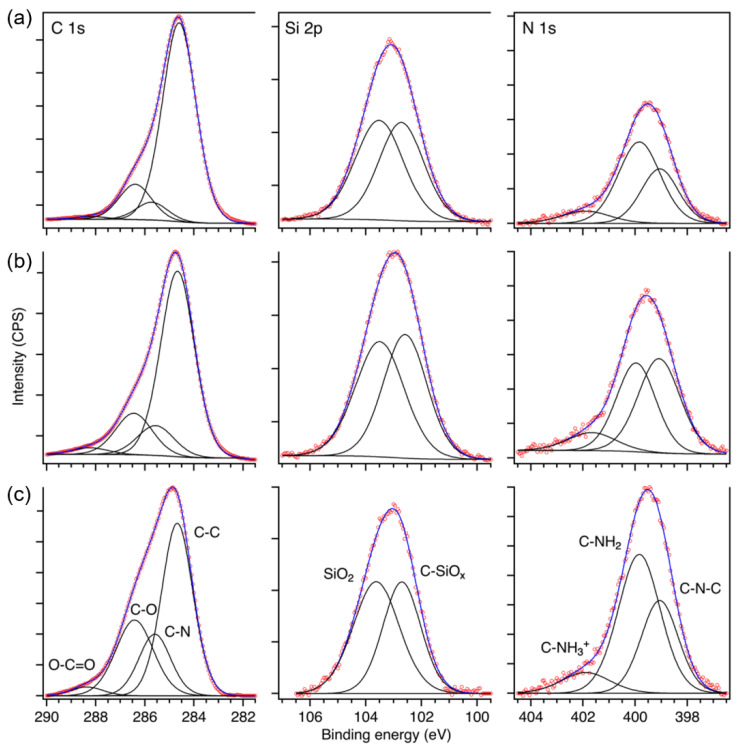
Deconvoluted C 1s, Si 2p, and N 1s XPS spectra of epoxy–silica coatings prepared with different DETA/DGEBA ratios: (**a**) D0.3, (**b**) D0.4, and (**c**) D0.6.

**Figure 4 polymers-14-03474-f004:**
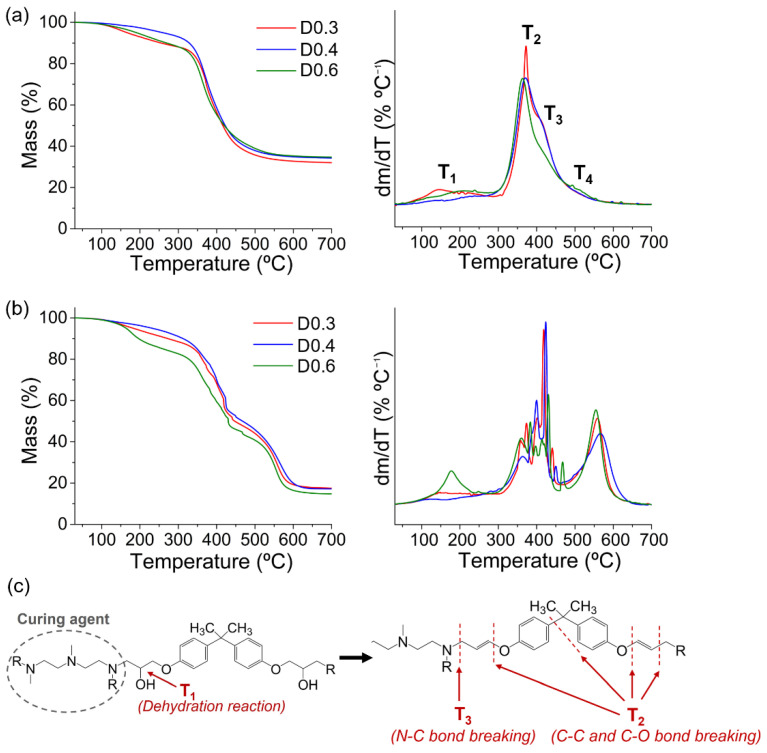
TG and DTG curves of epoxy–silica hybrids prepared with different DETA/DGEBA ratios in (**a**) nitrogen atmosphere, (**b**) air atmosphere, and (**c**) thermal decomposition mechanism of the hybrid structure in a nitrogen atmosphere, where R represents the bond with the DGEBA epoxy resin.

**Figure 5 polymers-14-03474-f005:**
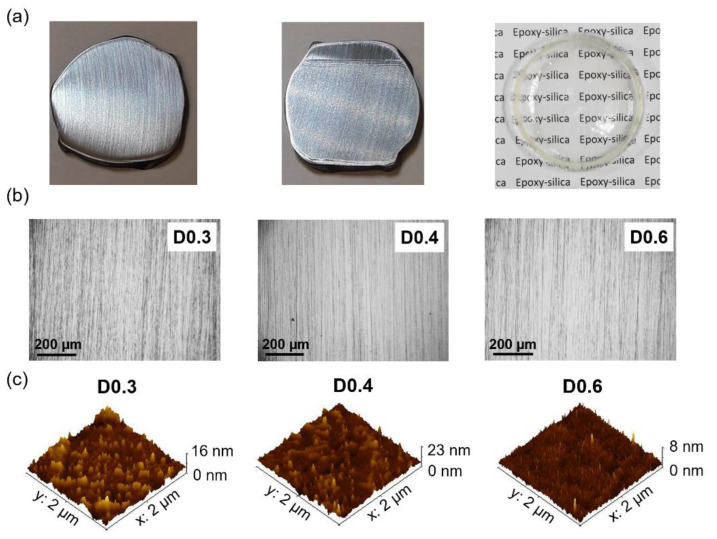
(**a**) Representative images of the uncoated reinforcing steel substrate (**left**), the epoxy-silica-coated steel (**center**), and the freestanding hybrid (**right**); (**b**) Optical micrographs of the coatings deposited on the reinforcing steel substrate (the dots and scratch marks are characteristic of the polished substrate); and (**c**) 3D AFM topography images of the coatings prepared with different proportions of the curing agent.

**Figure 6 polymers-14-03474-f006:**
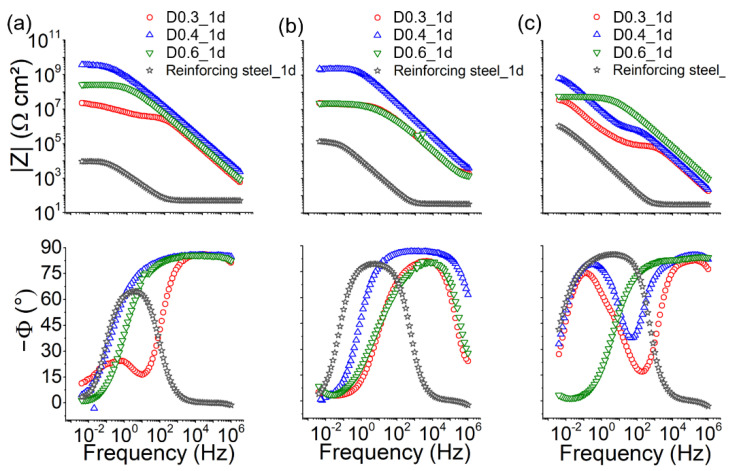
Bode plots of epoxy–silica hybrid coatings prepared with different DETA/DGEBA ratios after 1 day of exposure in solution: (**a**) 3.5 wt.% NaCl, (**b**) SCPS1 (pH 8), and (**c**) SCPS2 (pH 14).

**Figure 7 polymers-14-03474-f007:**
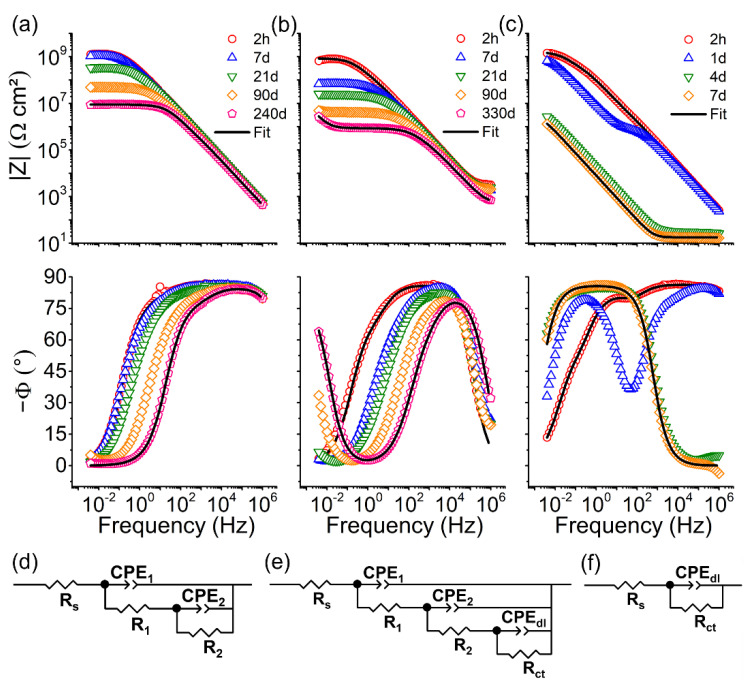
Bode plots for epoxy–silica hybrid coating prepared with DETA/DGEBA = 0.4 as a function of immersion time in: (**a**) 3.5 wt.% NaCl, (**b**) SCPS1 (pH 8), and (**c**) SCPS2 (pH 14) solution. (**d**–**f**) Equivalent electrical circuits (EEC) used to fit the EIS data.

**Figure 8 polymers-14-03474-f008:**
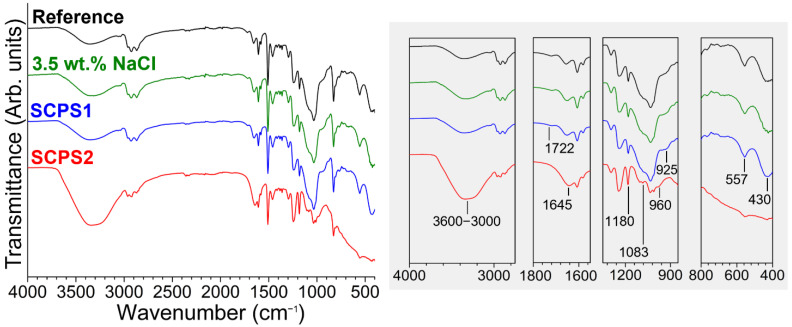
FTIR spectra of D0.4 freestanding films before and after 30 days of immersion in 3.5 wt.% NaCl, SCPS1, and SCPS2 solutions.

**Table 1 polymers-14-03474-t001:** Composition and pH of carbonated (SCPS1) and alkaline (SCPS2)-simulated concrete pore solutions.

Solution	Concentration (mol L^−1^)						pH
	Ca(OH)_2_	NaOH	KOH	Na_2_SO_4_	CaCO_3_	NaCl	
SCPS1	-	-	-	0.002	0.001	0.014	8
SCPS2	0.01	0.1	0.3	0.002	-	-	14

**Table 2 polymers-14-03474-t002:** Characteristic bands of epoxy–silica hybrids and precursor reagents.

Wavenumber (cm^−1^)	Attribution	Precursors	Refs.
3600–3080	OH of H−OH, Et−OH, and Si−OH/ N−H of primary and secondary amines	DETA	[24,28]
2995–2850	C−H of CH_2_, CH_3_/CH aromatic and aliphatic	DGEBA, DETA, GPTMS, TEOS	[23,24]
1722	C=O	−	[29]
1661	N−H of primary amines	DETA	[24]
1610	C=C of aromatic rings	DGEBA	[24]
1578	C=C of aromatic rings/N−H of primary amine	DGEBA, DETA	[24,25]
1508	C−C of aromatic	DGEBA	[24]
1458	CH_2_	DGEBA, DETA, GPTMS	[24]
1235	C−C−O−C	DGEBA	[26,27]
1180	Si−O−Si of ≡Si−O−Si≡/C−C−O−C	DGEBA, GPTMS, TEOS	[23,26,27]
1083	Si−O−Si of ≡Si−O−Si≡/C−N bond	DETA, GPTMS, TEOS	[23,27]
1035	C−O−C of ethers	DGEBA	[24]
915	C−O−C of oxirane group	DGEBA	[24]
825	1,4-aromatic ring substitution	DGEBA	[28]
806	Si−O of ≡Si−O−Si≡	−	[23]
557	Si−O of ((SiO)_4_) *	−	[23]
430	Si−O of O−Si−O	−	[23]

* Four-fold siloxane structures.

**Table 3 polymers-14-03474-t003:** Properties of the epoxy–silica hybrids: Porod exponent (*α*), dimension shape variable (*s*); and gyration radius (*R_g_)* determined by SAXS; adhesion strength to the steel substrate measured by pull-off test; surface roughness (*R_RMS_*) extracted by AFM; water contact angle; film thickness obtained by optical interferometry; porosity percentage obtained by pycnometry.

Sample	α	*s*	*R_g_* (nm)	Adhesion (MPa)	*R_RMS_* (nm)	Contact Angle (°)	Thickness (μm)	Porosity (%)
D0.3	3.0	0.32	10.0	6.5	1.7	78.3	9.4	3.8
D0.4	2.7	0.28	11.3	8.9	1.9	88.7	8.3	3.4
D0.6	2.9	0.48	9.6	16.1	0.6	82.2	15.2	3.1

**Table 4 polymers-14-03474-t004:** Characteristic events and parameters of thermal degradation of epoxy–silica hybrids, measured in duplicate (see text).

Sample	T_0_	T_1_	T_2_ in N_2_ (°C)	T_3_	T_4_	T_0_ in Air (°C)	Residue in N_2_ (%)	Residue in Air (%)
D0.3	254/309	143/190	362/373	413/411	-/-	272/214	32/33	17/18
D0.4	328/321	227/244	369/365	418/413	-/-	312/246	34/32	17/17
D0.6	269/267	207/204	363/358	417/402	513/500	196/187	34/30	14/15

**Table 5 polymers-14-03474-t005:** Impedance modulus (|Z|_lf_) after 1 day, |Z|_lf_ normalized to the thickness of 10 µm, and lifespan of epoxy–silica coatings on reinforcing steel recorded in duplicate during exposure to neutral saline and simulated concrete pore solutions.

	3.5 wt.% NaCl		SCPS1 (pH 8)		SCPS2 (pH 14)	
Sample	|Z|_lf_/|Z|_lf__10 µm (GΩ cm^2^)	Lifespan (Days)	|Z|_lf_/|Z|_lf__10 µm (GΩ cm^2^)	Lifespan (Days)	|Z|_lf_/|Z|_lf__10 µm (GΩ cm^2^)	Lifespan (Days)
D0.3	1.3/1.4	70	0.08/0.08	100	0.03/0.03	7
	0.02/0.02	60	0.02/0.02	60	0.02/0.02	7
D0.4	1.3/1.6	240	0.02/0.02	330	0.6/0.7	7
	3.8/4.6	76	1.9/2.3	70	0.07/0.08	7
D0.6	10.4/6.7	170	1.8/1.2	180	0.06/0.04	7
	0.3/0.2	60	0.02/0.01	28	0.06/0.04	7

**Table 6 polymers-14-03474-t006:** Electrochemical parameters obtained by fitting the EIS data of the D0.4 epoxy–silica coating after immersion in NaCl and SCPS solutions, using the EEC shown in Figure 6d–f.

	3.5 wt.% NaCl (Neutral)		SCPS1 (pH 8)		SCPS2 (pH 14)	
	2 h	240 Days	2 h	330 Days	2 h	7 Days
R_1_ (GΩ cm^2^)	0.11	7.65 × 10^−4^	0.14	2.62 × 10^−4^	9.45 × 10^−3^	-
Q_1_ (nΩ^−^^1^ cm^−^^2^ s^n^)	0.54	0.86	0.63	1.70	1.17	-
n_1_	0.96	0.94	0.96	0.91	0.96	-
C_1_ (nF cm^−2^)	0.48	0.56	0.56	0.85	0.96	-
R_2_ (GΩ cm^2^)	1.29	8.14 × 10^−3^	0.71	5.99 × 10^−4^	0.43	-
Q_2_ (nΩ^−^^1^ cm^−^^2^ s^n^)	0.38	2.71	0.91	20.2	0.69	-
n_2_	0.64	0.65	0.69	0.61	0.81	-
C_2_(nF cm^−2^)	0.26	0.35	0.75	1.31	0.52	-
R_ct_ (GΩ cm^2^)	-	-	-	1.00 × 10^6^	1.21	2.84 × 10^−3^
Q_dl_ (nΩ^−^^1^ cm^−^^2^ s^n^)	-	-	-	1.16 × 10^4^	2.78	2.15 × 10^4^
n_dl_	-	-	-	0.92	0.71	0.95
C_3_ (nF cm^−2^)	-	-	-	9.66 10^4^	4.61	2.62 × 10^4^
χ^2^	2.13 × 10^−3^	5.61 × 10^−4^	7.03 × 10^−3^	1.67 × 10^−4^	1.47 × 10^−4^	2.47 × 10^−3^
*φ (%)* *	-	3.63	-	9.28	-	- **

* Water uptake calculated using the Brasher–Kingsbury relation. ** Coating degradation.

## Data Availability

All raw and processed data will be provided on request by the corresponding author.

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
