# Peer review of "Structural Properties of Epoxy–Silica Barrier Coatings for Corrosion Protection of Reinforcing Steel"

_polymers, 2022, doi:10.3390/polym14173474_

Round 1

Reviewer 1 Report

This paper examined the effect of the epoxy resin/hardener ratio on the properties of DGEBA-GPTMS-TEOS-based epoxy-silica hybrids as potential coatings for corrosion protection of reinforcing steel. In general, the manuscript is well written, the experiments are elaborately detailed and results are clearly illustrated. The manuscript is therefore recommended for publication after MINOR revisions.

1.     The number of references can be substantially reduced. Cite only the most relevant references. Usually 30-40 references are sufficient for a journal article.

2.     Figure 4(a-d), the scale in horizontal coordinate could start from 0 so that the readers can easily identify and collect the data of the graphs.

3.       Conclusions should be restructured into four to five main points.

Author Response

This paper examined the effect of the epoxy resin/hardener ratio on the properties of DGEBA-GPTMS-TEOS-based epoxy-silica hybrids as potential coatings for corrosion protection of reinforcing steel. In general, the manuscript is well written, the experiments are elaborately detailed and the results are clearly illustrated. The manuscript is therefore recommended for publication after MINOR revisions.

Response: Thank you, we appreciate your positive response.

Point 1: The number of references can be substantially reduced. Cite only the most relevant references. Usually 30-40 references are sufficient for a journal article.

Response: Thank you for the suggestion. The references (page 18) were checked to maintain the most relevant. The number of references was reduced from 71 to 44, as suggested by the reviewer.

Point 2: Figure 4(a-d), the scale in horizontal coordinate could start from 0 so that the readers can easily identify and collect the data of the graphs.

Response: Thank you for this suggestion, however, as described in section 2.2.1 Structural and thermal properties (page 4):  the thermogravimetric (TG) analyses were performed for the samples in “…the temperature range from 30 to 700 °C”. Thus, the x-axis in Figure 4 a-b on page 11 is shown in the temperature range in which the analyses were performed:

Figure 4. TG and DTG curves of epoxy-silica hybrids prepared with different DETA/DGEBA ratios in a) nitrogen atmosphere, b) air atmosphere, and c) thermal decomposition mechanism of the hybrid structure in a nitrogen atmosphere, where R represents the bond with the DGEBA epoxy resin.

Point 3: Conclusions should be restructured into four to five main points.

Response: As suggested, we have partially rewritten and restructured the Conclusions (page 16) to five main points:

“Epoxy-silica hybrids were obtained by combining the curing reactions and sol-gel process using different ratios of curing agent and epoxy resin (DETA/DGEBA), in the range of 0.3-0.6, and deposited on a reinforcing steel substrate. Coatings with a thickness of a few micrometers exhibit a smooth and homogeneous surface and high adhesion to the reinforcing steel substrate of up to 16.1 MPa.

FTIR and XPS analyses confirmed the formation of the epoxy-silica hybrid structure.

Thermal and EIS analyses showed that an intermediate DETA to DGEBA molar ratio of 0.4 resulted in hybrid coatings with a modified nanostructure that provides high thermal stability (328 °C) and excellent long-term anticorrosive performance in saline and alkaline (pH 8) medium.  This was confirmed by SAXS and AFM results revealing the presence of a uniform distribution of silica domains with a size of about 10 nm covalently linked to the epoxy matrix, resulting in surface roughness of 1.9 nm and a contact angle of 88.7°.

 Long-term EIS assays of D0.4 coatings in 3.5 wt.% NaCl and SCPS1 (pH 8) solutions showed excellent durability of 330 days maintaining a high corrosion resistance, with a low-frequency impedance modulus of up to 3.8 GΩ cm2. The improved corrosion performance of the D0.4 coatings can be related to the presence of lager silica nanodomains filling the cross-linked structure of the epoxy resin, acting as an efficient barrier against electrolyte permeation. A similar initial performance was observed in the SCPS2 solution (pH 14), however, with shorter durability due to coating degradation induced by hydrolysis reactions of DGEBA resin and the silica nanophase in a harsh environment.

The results presented here highlight the potential of epoxy-silica coatings for the protection of reinforcing steel in aged carbonated (pH 8) concrete structures and point to their limitations and harsh (pH 14) alkaline electrolytes.”

Reviewer 2 Report

The manuscript deals with the effect of epoxy resin as a coating for steel reinforcement to protect from corrosion. According to the reviewer's opinion, the manuscript is well-structured and clear. The topic is interesting and very important for the reinforced concrete structures exposed to severe environmental conditions. In addition, the results are well-presented and could be helpful to the further development of a similar topic. Hence, the paper can be accepted in its current form.

Author Response

Response: Thank you very much for your positive comments.
